# Relational Diffusion Distillation For Efficient Image Generation

## ABSTRACT

Although the diffusion model has achieved remarkable performance in the field of image generation, its high inference delay hinders its wide application in edge devices with scarce computing resources. Therefore, many training-free sampling methods have been proposed to reduce the number of sampling steps required for diffusion models. However, they perform poorly under a very small number of sampling steps. Thanks to the emergence of knowledge distillation technology, the existing training scheme methods have achieved excellent results at very low step numbers. However, the current methods mainly focus on designing novel diffusion model sampling methods with knowledge distillation. How to transfer better diffusion knowledge from teacher models is a more valuable problem but rarely studied. Therefore, we propose Relational Diffusion Distillation (RDD), a novel distillation method tailored specifically for distilling diffusion models. Unlike existing methods that simply align teacher and student models at pixel level or feature distributions, our method introduces cross-sample relationship interaction during the distillation process and alleviates the memory constraints induced by multiple sample interactions. Our RDD significantly enhances the effectiveness of the progressive distillation framework within the diffusion model. Extensive experiments on several datasets (e.g., CIFAR-10 and ImageNet) demonstrate that our proposed RDD leads to 1.47 FID decrease and 256x speed-up, compared to state-of-the-art diffusion distillation methods. Our code will be attached to the supplementary material.

## CCS CONCEPTS

• **Computing methodologies** → **Computer vision**.

## KEYWORDS

Diffusion models, Relational distillation, Progressive distillation

## 1 INTRODUCTION

Recently, generative artificial intelligence has attracted more and more attention. Generative AI is a special type of AI algorithm that, unlike discriminative AI algorithms, focuses more on generating new content such as text [4], images [29], or even videos [1]. Focusing on the field of image generation, the information density of images is much lower than that of text, thus the difficulty of generating high-quality images is much higher than that of generating high-quality text content. Benefiting from the research of basic models in the field of image generation in recent years, more

*ACM MM, 2024, Melbourne, Australia*

© 2024 Copyright held by the owner/author(s). Publication rights licensed to ACM.
ACM ISBN 978-x-xxxx-xxxx-x/YY/MM
https://doi.org/10.1145/nnnnnnn.nnnnnnn

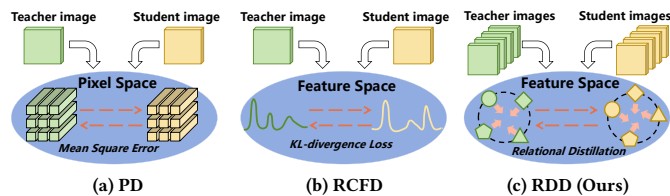

**Figure 1: Different distillation targets between (a) PD, (b) RCFD, and (c) our proposed RDD.**

and more powerful generative models have been proposed to solve the problem of image generation, such as GAN [8]. However, GAN models suffer from training difficulties, and the model architecture and some hyperparameters of the model need to be carefully designed. The diffusion model [14, 28, 34] not only overcomes these difficulties but also achieves better performance with its excellent generation quality [5], which makes it possible to generate high-quality images at higher resolutions.

However, the remarkable generative capacity of the diffusion model mainly stems from its iterative denoising procedure [14]. Thus, the extensive iteration required for generation inherently slows down its inference pace compared to the GAN model, which necessitates only a single inference step. This sluggish inference rate of the diffusion model presents obstacles to its deployment on edge devices with limited computational resources and its broader application. However, directly reducing the number of sampling steps of the diffusion model can lead to serious performance degradation [34]. Thus, enhancing the inference speed of the diffusion model while preserving its generative prowess to the utmost degree emerges as a profoundly significant challenge. To mitigate the number of sampling steps required by the diffusion model, two primary approaches have been proposed: training-free sampling and training schemes [2]. Within these approaches, the training-free method [24, 26, 34] endeavors to devise more efficient sampling techniques to expedite inference, while the training scheme method [19, 28, 33, 35] necessitates the incorporation of an additional training phase. Despite introducing an extra training process, the training scheme offers the potential for diffusion models to excel with remarkably few sampling steps (1-8 steps) [33, 35].

Recently, training schemes leveraging knowledge distillation [33, 35] have yielded remarkable results with an exceedingly low number of sampling steps, surpassing the performance of other methods [19, 24, 26, 28]. Knowledge distillation, as proposed by Hinton et al. [13], aims to distill knowledge from a more robust yet larger teacher model to craft a streamlined student model that inherits the teacher's superior performance to the greatest extent possible. These distillation-based methods can generally be categorized into two groups: consistency distillation [35] and progressive distillation [33]. Consistency distillation employs the guidance of the teacher model to distill a student model with a minimal number of sampling steps within a single training iteration, albeit with a

lengthier training duration. On the other hand, progressive distillation achieves a student model with half the sampling steps in each training iteration under the guidance of the teacher model. Through multiple training sessions, a student model with remarkably few sampling steps is attained. By employing the aforementioned distillation framework anchored on a potent teacher model, we can ultimately derive a student model with a minimal number of sampling steps while maintaining commendable performance, significantly accelerating the inference speed of the diffusion model.

The prevailing accelerated sampling distillation methods for diffusion models primarily center on refining the distillation framework itself. This entails devising more effective sampling strategies for the student network, leveraging insights from the teacher model to train the student network with minimal sampling steps. Consistency distillation [35] aims to directly train a student model with a reduced number of sampling steps, while progressive distillation [33] endeavors to halve the sampling steps of the student model with each training iteration. However, following the formulation of the distillation strategy for the student network, the challenge of transferring the abundant knowledge from the teacher model often remains overlooked. Existing methods [33] typically focus solely on aligning the teacher and student models at the pixel level, neglecting the semantic information inherent in image generation tasks. Approaches based on image features [35, 36] typically employ direct feature alignment, without delving into whether more intricate features could enhance performance further, or how to construct features that facilitate easier learning from diffusion models.

In detail, in knowledge distillation within diffusion models, PD [33] utilizes Mean Square Error (MSE) to directly align images at the pixel level. CM [35] employs both MSE and Learned Perceptual Image Patch Similarity (LPIPS) [44] to gauge image disparities, revealing LPIPS as notably superior to MSE. RCFD [36] integrates an additional feature extractor based on PD to compute the KL divergence between features for achieving fine-grained alignment through image feature extraction. Among these approaches, LPIPS utilizes a pre-trained network to extract feature maps from different layers for direct alignment, while CM does not explore a distillation knowledge format more suited to the diffusion model. RCFD relies solely on KL divergence for alignment, which inherently sacrifices valuable spatial information within feature maps. Additionally, for different images, the features between them are naturally different, and the aforementioned distillation methods incorporating features only consider the alignment between individual samples, potentially leading to suboptimal optimization outcomes. Inspired by the feature distillation method of diffusion models, we enhance the training framework of RCFD and introduce a distillation approach termed **R**elational **D**iffusion **D**istillation (**RDD**) within the progressive distillation framework of the diffusion model. Fig. 1 shows the distillation target of the different methods. Firstly, we introduce **I**ntra-**S**ample **P**ixel-to-**P**ixel Relationship Distillation (IS_P2P), wherein we construct pairwise spatial relation matrices to retain spatial information within feature maps. Moreover, cross-sample relationship interaction is introduced to capture long-term dependencies between image features. Subsequently, we propose **M**emory-based **P**ixel-to-**P**ixel Relationship Distillation (M_P2P). By establishing an online pixel queue, consistent contrastive embeddings are obtained from past samples,

enabling the calculation of a pixel similarity matrix. This approach resolves the memory inefficiency associated with multiple sample interactions and introduces a greater diversity of features through the inclusion of more contrastive embeddings.

In this paper, we contribute to the advancement of progressive diffusion model distillation by integrating feature map spatial information and establishing information interaction pathways between samples. Our key contributions can be outlined as follows:

- We introduce a novel distillation method tailored specifically for diffusion models, termed Relational Diffusion Distillation (RDD). This method significantly enhances the effectiveness of the progressive distillation framework within the diffusion model.
- We propose the Inter-Sample Pixel-to-Pixel Relationship Distillation, leveraging spatial information embedded within feature maps. This method introduces cross-sample relationship interaction during the distillation process, enhancing knowledge transfer across samples.
- We introduce the Memory-based Pixel-to-Pixel Relationship Distillation, which utilizes memory to establish an online queue. This approach alleviates the memory constraints induced by multiple sample interactions, while simultaneously enhancing the diversity of samples and amplifying direct information interaction between students and teachers.
- We conduct a thorough ablation study on the proposed Relational Diffusion Distillation to affirm the efficacy of the introduced techniques. Through comprehensive evaluation, we demonstrate that our Relational Diffusion Distillation outperforms the existing Classifier-based Feature Distillation method.

## 2 RELATED WORK

**Diffusion Model.** A well-trained diffusion model can obtain high-quality generated images by denoising random Gaussian noise step by step, and its standard training process was first proposed in DDPM [14]. In the inference phase, for a diffusion model with parameter $\theta$, it can take a noisy image $\mathbf{z}_t$ and a time $0 \leq t \leq 1$ as inputs and outputs a denoised image $\mathbf{x}_t = \theta(\mathbf{z}_t, t)$. By starting from $t = 1$, the denoised process is repeated $N$ times to get the final image, where $N$ is the sampling steps of the trained diffusion model. Usually, $N$ is a relatively large number(e.g., 512, 1024), and the inference process is time-consuming. Thus DDIM [34] proposes an implicit sampling to speed up the inference process by the following equation.

$$\mathbf{z}_s = \alpha_s \theta(\mathbf{z}_t, t) + \sigma_s \frac{\mathbf{z}_t - \alpha_t \theta(\mathbf{z}_t, t)}{\sigma_t} \qquad (1)$$

where $\alpha$ and $\sigma$ are pre-defined time correlation coefficients, and $0 \leq s < t \leq 1$. When $t = 1$, $\mathbf{z}_t$ is a standard gaussian noise, and $\mathbf{z}_s$ is the final image when $s = 0$.

**Knowledge Distillation.** Knowledge Distillation facilitates the creation of a superior student model by transferring knowledge from a larger, more advanced teacher model to a more compact student model. Since the seminal work by Hinton et al. [13] introduced the use of KL divergence to distill model logits, many Knowledge Distillation methods have emerged to address various challenges. In contrast to logits-based Knowledge Distillation, there's a growing

recognition that the intermediate feature layers within a network also harbor valuable information, which can serve as guidance for the student model's learning process. Consequently, feature-based Knowledge Distillation techniques have been devised. For instance, Fitnet [30] leverages the intermediate feature layer of the network to transfer knowledge from the teacher network, while AT [42] aggregates the intermediate feature layer across channel dimensions to derive attention maps as knowledge. Beyond directly learning features, some approaches utilize the relationships between multiple feature maps as knowledge to guide the student model. For instance, DGB [18] focuses on learning the relationship between global and local features of the teacher network. These Knowledge Distillation techniques find applications in diverse domains such as image classification [9, 10, 17, 22, 47], object detection [21, 40, 43, 46], image segmentation [6, 16, 25, 39, 41], and beyond, yielding remarkable outcomes. However, despite their widespread adoption, no prior research has explored the application of these advanced distillation techniques in the context of distilling diffusion models.

**Diffusion Acceleration.** Improving the speed of generation in the diffusion model stands as a perennially critical challenge. The DDIM [34] dynamically tunes the sampling step size by mitigating random noise from DDPM. This adjustment notably diminishes the requisite sampling steps while maintaining a comparable generation quality, albeit displaying suboptimal results at very low sampling steps. On the other hand, PD [33] leverages a teacher model to mentor the student model, enabling the latter's single sampling to approximate the quality of the former's double sampling, thereby progressively halving the sampling steps. Additionally, RCFD [36] integrates supplementary image classifiers to extract features from images generated in PD, supplanting pixel-level MSE as a novel optimization objective. CM [35] attains minimal step generation by directly learning from the raw data distribution. Furthermore, Snap [23] crafts a model with fewer parameters yet yields superior effects to diminish the delay of a single inference. Mobile diffusion [45] achieves comparable generation effects with reduced computation by refining the infrastructure of the diffusion model. Nonetheless, these methodologies mainly concentrate on enhancing diffusion model distillation architectures, with limited exploration of specific diffusion knowledge forms. This potentially leads to sub-optimal distillation performance. Hence, this paper primarily delves into the design of a meticulous distillation technique tailored for the diffusion model, aimed at better aligning the generated image details and realizing an enhanced distillation effect.

## 3 PRELIMINARY

Firstly, we introduce Progressive Distillation (PD) [33] and Classifier-based Feature Distillation (CFD) [36].

### 3.1 Progressive distillation

Based on DDIM, Progressive Distillation accelerates the sampling process of the diffusion model by the knowledge distillation method. Assuming that there is now a well-trained teacher model with $N$ sampling steps, we can use PD to train a student model with parameter $\theta$ and $N/2$ sampling steps. Formally speaking, given a sampling time $t$ and a noisy image $\mathbf{z}_t$, the denoised image $\mathbf{x}^T$ at time $t - 2/N$ can be generated by the teacher model. The detailed

derivation for $\mathbf{x}^T$ is provided in Appendix. Then, we can calculate the training loss for the student model by

$$\mathcal{L}_{PD} = \omega_t ||\mathbf{x}^T - \theta(\mathbf{z}_t, t)||_2^2 \tag{2}$$

where $\omega_t = max(\frac{\alpha_t^2}{\sigma_t^2}, 1)$ is used for better performance.

### 3.2 Classifier-based feature distillation

In PD, Mean Square Error (MSE) serves as the metric for aligning images generated by the teacher and student models at the pixel level. In contrast, Classifier-based Feature Distillation (CFD) adopts an alternative approach by incorporating an additional feature extractor to align images based on feature dimensions. At RCFD [36], a pre-trained classifier is employed as the feature extractor. This classifier, denoted as $cls$, consists of two components: the feature extractor $extr$ and fully connected layers.

Formally, when presented with an image $\mathbf{x}$, the feature extractor $extr$ operates to extract features, yielding $\mathbf{F} = extr(\mathbf{x})$. In CFD, solely the feature information is utilized, disregarding the fully connected layers. Consequently, instead of directly assessing the images $\mathbf{x}^T$ and $\mathbf{x}^S = \theta(\mathbf{z}_t, t)$ generated by the teacher and student models, respectively, the extractor $extr$ is employed to extract features, expressed as:

$$\mathbf{F}^T = extr(\mathbf{x}^T), \mathbf{F}^S = extr(\mathbf{x}^S) \tag{3}$$

After this, we can obtain the feature distribution by using the softmax function $\sigma(\cdot)$ and calculate the KL-divergence between teacher and student image feature distributions

$$\mathcal{L}_{CFD} = \text{KL}\big(\sigma(\mathbf{F}^T/\tau), \sigma(\mathbf{F}^S)\big) \tag{4}$$

where $\tau$ is a pre-defined temperature to soften teacher distribution for a better distillation process. RCFD [36] found that softening only the teacher distribution has a better effect. As image features often have more information than image pixels, by using the training framework of PD and replacing the $\mathcal{L}_{PD}$ with $\mathcal{L}_{CFD}$, better image generation quality is achieved in RCFD [36].

## 4 METHOD

The triumph of CFD underscores that within the PD framework, aligning feature dimensions between images surpasses mere pixel-level alignment. This superiority stems from the enriched semantic information encapsulated within the features extracted by $extr$, facilitating the acquisition of robust visual representations during the distillation process. Nevertheless, the efficacy of this approach prompts a pertinent question: is employing KL divergence alone adequate for feature alignment, and could leveraging image features to construct distillation information enhance the learning process within the diffusion model?

Reviewing the formula for computing KL divergence, when presented with two distributions $q$ from the teacher model and $p$ from the student model, the KL loss can be determined as follows:

$$\mathcal{L}_{KL}(q||p) = \sum_i q_i log \frac{q_i}{p_i} \tag{5}$$

The essence of the loss function aims to minimize the disparity between the distributions of students and teachers. However, when dealing with feature tensors $\mathbf{F}^T$ and $\mathbf{F}^S$ of dimensions $\mathbb{R}^{H \times W \times C}$,

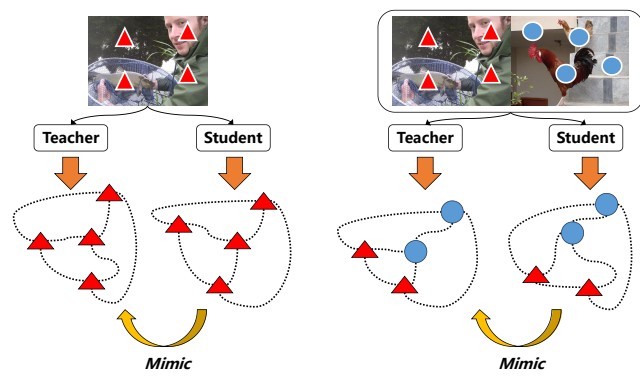

(a) Intra-image distillation     (b) Intra-sample distillation

**Figure 2: Difference between Intra-image and Intra-sample pixel-to-pixel distillation.**

computing the KL loss in RCFD [36] necessitates using average pooling which makes features into $\mathbb{R}^C$ and subsequently applying the softmax function to derive the feature distribution. Regrettably, this process results in the complete loss of spatial information embedded within the features. In the context of image generation, spatial information is vital because features across different locations may exhibit correlations. For instance, adjacent features within a single object tend to be more akin, whereas those at the object's boundaries often display greater disparity. Therefore, it becomes imperative to incorporate spatial information into the learning process to enhance the distillation performance. Consequently, we hope to integrate spatial information into the feature distillation process in an effective way.

## 4.1 Intra-Sample Pixel-to-Pixel Relationship Distillation

In the distillation process, to retain the spatial information of the feature map, we use the last convolutional layer output $\mathbf{F} \in \mathbb{R}^{H \times W \times C}$ of *extr* instead of the average pooling feature map. For $\mathbf{F} \in \mathbb{R}^{H \times W \times C}$ we firstly preprocess it by $l_2$-normalization and for easy notation, we reshape the spatial dimension into $\mathbf{F} \in \mathbb{R}^{A \times C}$, where $A = H \times W$. Subsequently, the spatial relation matrix $\mathbf{M} = \mathbf{F}\mathbf{F}^{\mathsf{T}} \in \mathbb{R}^{A \times A}$ is computed. This matrix encapsulates the spatial relationships between pixels, denoted as $\mathbf{M}^T$ and $\mathbf{M}^S$, thus ensuring the retention of spatial information within the feature map. Termed Intra-Image Pixel-to-Pixel Relationship Distillation (II_P2P), this approach enables the teacher model to utilize knowledge distillation to guide image generation in the student model by leveraging the relative relationships between pixels. The distillation process is thus formulated as follows:

$$\mathcal{L}_{II\_P2P}(\mathbf{M}^T, \mathbf{M}^S) = \frac{1}{A} \sum_{a=1}^{A} \mathrm{KL}\big(\sigma(\frac{\mathbf{M}_{a,:}^T}{\tau}), \sigma(\frac{\mathbf{M}_{a,:}^S}{\tau})\big) \quad (6)$$

where $\tau$ is a pre-defined temperature to soften distribution for a better distillation process. We use the softmax function to mitigate the magnitude gaps between the two models and use KL-divergence loss to align row-wise probability distribution.

However, II_P2P solely accounts for the spatial relationships within individual images when computing the spatial relation matrix. Consequently, only the spatial information of a single sample is modeled, failing to capture broader contextual insights. For the task of image generation, a mature teacher model possesses the capability to generate diverse images exhibiting distinct features (e.g., cats, dogs). Consequently, when constructing the relation matrix, it becomes imperative to consider not only the spatial relationships within individual images but also the interplay between multiple images. By doing so, a single pixel feature can engage with a broader array of images, thereby enabling the student model to capture long-term dependency relationships between image features. This approach, distilled from a mature teacher model, facilitates the enhancement of the model's visual representation abilities and ultimately elevates the quality of image generation.

Hence, we introduce Intra-Sample Pixel-to-Pixel Relationship Distillation (IS_P2P). Given a mini-batch sample $\{\mathbf{x}_n\}_{n=1}^N$ generated by diffusion models, the extraction of features by *extr* yields $N$ feature maps denoted as $\{\mathbf{F}_n \in \mathbb{R}^{A \times C}\}_{n=1}^N$. It's worth noting that these features are reshaped akin to the operations in II_P2P. For the $i$-th sample $\mathbf{x}_i$ and the $j$-th sample $\mathbf{x}_j$, with $i, j \in \{1, 2, \cdots, N\}$, we compute pair-wise spatial relation matrices $\mathbf{R}_{i,j} = \mathbf{F}_i \mathbf{F}_j^{\mathsf{T}} \in \mathbb{R}^{A \times A}$. Consequently, $\mathbf{R} \in \mathbb{R}^{N \times N \times A \times A}$ embodies the mini-batch intra-sample spatial relation matrix. We utilize pair-wise spatial relation matrices $\mathbf{R}_{i,j}^T$ from the teacher model to guide those of $\mathbf{R}_{i,j}^S$ from the student model. Fig. 2 shows the difference between our proposed II_P2P and IS_P2P. We also compare their performance in section 5.3. The distillation process is thus formulated as follows:

$$\mathcal{L}_{IS\_P2P} = \frac{1}{N^2} \sum_{i=1}^{N} \sum_{j=1}^{N} \mathcal{L}_{II\_P2P}(\mathbf{R}_{i,j}^T, \mathbf{R}_{i,j}^S) \quad (7)$$

The overview of our proposed IS_P2P is shown in Fig. 3.

## 4.2 Memory-based Pixel-to-Pixel Relationship Distillation

While IS_P2P effectively captures relational features among multiple pairs and facilitates interactions among sample features within each mini-batch, it exhibits certain limitations. Notably, IS_P2P solely encompasses samples within each mini-batch, thereby overlooking the diversity of features across different mini-batches. Moreover, to enhance feature diversity in IS_P2P, smaller batch size is undesirable as it restricts the number of pair-wise spatial relation matrices and consequently diminishes feature diversity. Conversely, a larger batch size, although beneficial for feature diversity, proves to be hardware-unfriendly due to its extensive memory requirements and we verify this in section 5.3. To address this problem, we propose a memory-based pixel queue capable of storing a vast array of distinct pixel embeddings from past samples terms as Memory-based Pixel-to-Pixel Relationship Distillation(M_P2P). Leveraging this pixel queue enables efficient storage and retrieval of numerous pixel embeddings from diverse samples, thereby mitigating the shortcomings of IS_P2P.

The concept of a memory bank was initially introduced within the field of self-supervised learning [37, 38]. In the context of self-supervised contrastive learning, the construction of a sizable pool

![Figure 3](left figure)

**Figure 3: Overview of Intra-Sample Pixel-to-Pixel Relationship Distillation.**

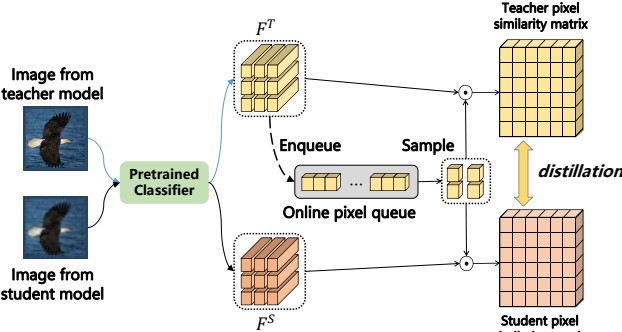

**Figure 4: Overview of Memory-based Pixel-to-Pixel Relationship Distillation**

of negative samples is imperative to ensure effective learning [11]. This aligns seamlessly with our requirements, as relying solely on a single batch of data is insufficient. When establishing the pixel queue, we must consider both memory constraints and the likelihood of redundancy among adjacent pixels in the feature map. To maximize the storage capacity for sample features while minimizing memory costs, we propose the creation of an online pixel queue denoted as $\mathbf{Q} \in \mathbb{R}^{N_q \times C}$, where $N_q$ represents the number of pixel embeddings and $C$ denotes the embedding dimension. For each image, we sample a small subset of pixel embeddings (denoted by $K$, where $K \ll N_q$) from the feature map and append them to the pixel queue. The updating mechanism for the queue adheres to the "first in, first out" strategy, ensuring the continual refreshment of stored pixel embeddings.

Drawing inspiration from [7], we propose the integration of a shared pixel queue between the teacher and student models, wherein pixel embeddings within the queue are generated by the teacher model during the distillation phase. Given $l_2$-normalized feature maps $\mathbf{F}_n^T$ and $\mathbf{F}_n^S \in \mathbb{R}^{A \times C}$ generated by the teacher and student models, respectively, we randomly sample $V$ pixel embeddings denoted as $\{\mathbf{e}_i \in \mathbb{R}^C\}_{i=1}^V$ from the pixel queue. Subsequently, we concatenate these embeddings into a matrix $\mathbf{E} = [\mathbf{e}_1, \mathbf{e}_2, \cdots, \mathbf{e}_V] \in \mathbb{R}^{V \times C}$. Thus we can compute the pixel similarity matrix between the feature maps as anchors and the pixel embeddings as contrastive embeddings.

$$\mathbf{P}^T = \mathbf{F}_n^T \mathbf{E}^\top \in \mathbb{R}^{A \times V}, \mathbf{P}^S = \mathbf{F}_n^S \mathbf{E}^\top \in \mathbb{R}^{A \times V} \quad (8)$$

In this way, the features of the student model interact directly with the features of the teacher model, and the gap between students and teachers is further smoothed by imitating the pixel similarity matrix of the teacher model. Similar to II_P2P, we use the softmax function to normalize row-wise distribution and use KL-divergence loss to perform pixel-to-pixel distillation. The memory-based distillation process is thus formulated as follows:

$$\mathcal{L}_{M\_P2P} = \frac{1}{A} \sum_{a=1}^A \mathrm{KL}\big(\sigma(\frac{\mathbf{P}_{a,:}^T}{\tau}), \sigma(\frac{\mathbf{P}_{a,:}^S}{\tau})\big) \quad (9)$$

where $\tau$ is a pre-defined temperature to soften distribution. Subsequently, after each iteration, we randomly select $K$ pixel embeddings from $\mathbf{F}_n^T$ and push them into the pixel queue $\mathbf{Q}$. The overview

of our proposed M_P2P is shown in Fig. 4. It's worth noting that while previous unsupervised learning methods [11] encountered training difficulty due to inconsistencies between anchors and contrastive embeddings, our task setting alleviates this concern. Since the teacher model is well-trained and kept frozen, all contrastive embeddings generated during the training process remain consistent. Therefore, the incorporation of additional training techniques is unnecessary, as it does not lead to training difficulty.

### 4.3 Overall Framework

We consolidate our Intra-Sample Pixel-to-Pixel Relationship Distillation and Memory-based Pixel-to-Pixel Relationship Distillation methodologies to train our student network. Additionally, we incorporate $\mathcal{L}_{CFD}$ as the fundamental loss. The overall loss of Relational Diffusion Distillation is formulated as:

$$\mathcal{L}_{RDD} = \mathcal{L}_{CFD} + \alpha \mathcal{L}_{IS\_P2P} + \beta \mathcal{L}_{M\_P2P} \quad (10)$$

where $\alpha$ and $\beta$ are weights coefficients. Although $\mathbf{F}_n^T$ and $\mathbf{F}_n^S$ possess the same embedding dimension owing to the shared pretrained classifier, we draw inspiration from [27] to enhance performance. Consequently, we append a projection head to $\mathbf{F}_n^S$ before the computation of $\mathcal{L}_{M\_P2P}$. This projection head comprises two 1×1 convolutional layers with ReLU activation and batch normalization. The projection head is discarded during the inference phase without incurring additional costs.

## 5 EXPERIMENT

### 5.1 Experimental Setup

**Dataset.** We validate the effectiveness of our method using the CIFAR-10 [20] dataset for unconditional generation and the ImageNet 64×64 [3] dataset for conditional generation.

**Evaluation metrics.** We report the Inception Score (IS) [32] and Fréchet Inception Distance (FID) [12] results of each method. IS measures the class balance and confidence of the generated images, while FID measures the difference in feature distribution between the generated and real images. Therefore, higher IS and lower FID represent better generated images.

**Network architectures.** We use the same network architectures used in RCFD [36] and we slightly modify it to fit the resolution of

| Sampling Steps | Method | IS ↑ | FID ↓ |
|---|---|---|---|
| 1 | PD[33] | 7.88 | 15.06 |
| | PD[33]+LPIPS[44] | 8.51 | 8.95 |
| | RCFD[36] | 8.87 | 8.92 |
| | **RDD** | **8.95** | **8.16** |
| 2 | PD[33] | 8.70 | 7.42 |
| | PD[33]+LPIPS[44] | 8.90 | 5.70 |
| | RCFD[36] | **9.19** | 5.07 |
| | **RDD** | 9.17 | **4.78** |
| 4 | PD[33] | 9.04 | 4.83 |
| | PD[33]+LPIPS[44] | 9.11 | 4.45 |
| | RCFD[36] | 9.34 | 3.80 |
| | **RDD** | **9.35** | **3.73** |
| 8 | PD[33] | 9.14 | 4.14 |
| | DDIM[34] | 8.14 | 20.97 |
| 10 | PNDMs[24] | - | 7.05 |
| 12 | DPM-Solver[26] | - | 4.65 |
| 1024 | DDIM[34] | 9.21 | 3.78 |

**Table 1: Performance comparison with other methods on CIFAR-10.**

| Sampling Steps | Method | IS ↑ | FID ↓ |
|---|---|---|---|
| 1 | PD[33] | 18.87 | 16.88 |
| | PD[33]+LPIPS[44] | 19.63 | 14.59 |
| | RCFD[36] | 22.88 | 13.44 |
| | **RDD** | **23.12** | **11.97** |
| 2 | PD[33] | 19.94 | 12.81 |
| | PD[33]+LPIPS[44] | 20.49 | 11.23 |
| | RCFD[36] | 23.20 | 9.54 |
| | **RDD** | **23.23** | **8.90** |
| 4 | PD[33] | 21.09 | 9.44 |
| | PD[33]+LPIPS[44] | 21.13 | 9.46 |
| | RCFD[36] | 22.63 | 8.08 |
| | **RDD** | **22.81** | **7.92** |
| 8 | PD[33] | 21.39 | 8.80 |
| | DDIM[34] | 19.35 | 20.72 |
| 128 | DDIM[34] | 21.02 | 8.95 |
| 1024 | DDIM[34] | 21.65 | 8.46 |

**Table 2: Performance comparison with other methods on ImageNet 64×64.**

ImageNet 64×64, details are provided in Appendix. We use the U-Net [31] as the diffusion model. DenseNet201 [15] as the classifiers and we pretrain it on both datasets.

**Hyper-parameters setting.** For the CIFAR-10 dataset, we set $\alpha = 1$ and $\beta = 0.1$ in the overall loss and for $\tau$ used in $\mathcal{L}_{CFD}$, we adhere to the settings outlined in the original paper [36], with $\tau_{8to4} = 0.9$, $\tau_{4to2} = 1.0$, and $\tau_{2to1} = 0.85$. For the ImageNet 64×64 dataset, we set $\alpha = 100$ and $\beta = 0.1$ and for $\tau$ used in $\mathcal{L}_{CFD}$, we set it as $\tau = 0.85$ for all experiment. In both cases, the distillation temperature $\tau$ for $\mathcal{L}_{IS\_P2P}$ is set to 1 and for $\mathcal{L}_{M\_P2P}$ we set it as 0.1. The pixel queue size $N_q$ is set to 20,000, and the pixel queue sample size $V$ is set to 2048.

| Loss | PD | RCFD | Distillation Loss | | | |
|---|---|---|---|---|---|---|
| $\mathcal{L}_{CFD}$ | - | ✓ | ✓ | ✓ | ✓ | ✓ |
| $\mathcal{L}_{II\_P2P}$ | - | - | ✓ | - | - | - |
| $\mathcal{L}_{IS\_P2P}$ | - | - | - | ✓ | - | ✓ |
| $\mathcal{L}_{M\_P2P}$ | - | - | - | - | ✓ | ✓ |
| FID | 15.06 | 8.92 | 8.45 | 8.35 | 8.47 | **8.16** |

**Table 3: Ablation study of distillation loss terms on CIFAR-10.**

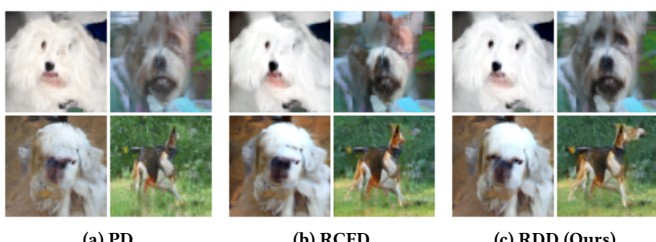

(a) PD      (b) RCFD      (c) RDD (Ours)

**Figure 5: Samples generated in one step by (a) PD, (b) RCFD, and (c) our proposed RDD on ImageNet 64×64. All corresponding images are generated from the same initial noise.**

**Training setting.** Following the configuration used in RCFD [36], we commence by distilling a basic model using Progressive Distillation (PD) from 1024-step to 8-step. Then we focus on the distillation process starting from 8-step to 1-step with different methods. The detailed training parameters can be found in Appendix.

**Compared distillation methods.** We compare our proposed Relational Diffusion Distillation with training-free sampling methods DDIM [34], PNDMs [24], DPM-Solver [26] and training scheme methods PD [33] and RCFD [36] in CIFAR-10. We compare with DDIM [34], PD [33] and RCFD [36] in ImageNet 64×64. As mentioned above, we also compare our method with PD with LPIPS [44] so that we can better compare different feature-based distillation methods. We re-run all methods based on the code provided by RCFD.

## 5.2 Experimental Results

**Results on CIFAR-10.** In Table 1, we compare our proposed RDD method for unconditional generation on CIFAR-10 with other mentioned methods. We observe that compared to RCFD, RDD yields FID reduces of 0.76, 0.29, and 0.07 at 1, 2, and 4 steps sampling, while maintaining or even improving the Inception Score (IS), which signifies a notable advancement over the PD method. Moreover, RDD demonstrates superior performance with a reduced number of sampling steps, indicating its potential for highly efficient distillation experiments with minimal steps. Notably, our RDD method at 4 sampling steps even surpasses DDIM at 1024 steps with a remarkable 256× increase in sampling speed. Furthermore, RDD achieves comparable generation quality to DPM-Solver's 12-step sampling with only 2-step sampling, effectively reducing the number of sampling steps by 6×. These experimental results underscore the superior or equivalent performance achieved by RDD in fewer sampling steps

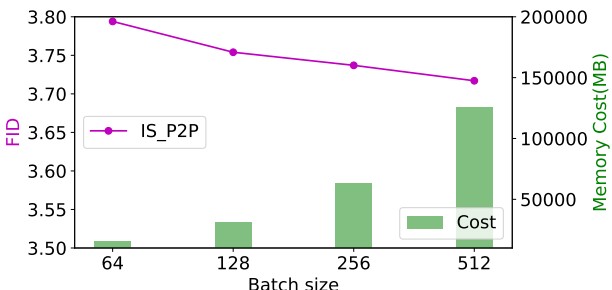

**Figure 6: Impact of different batchsize with $\mathcal{L}_{IS\_P2P}$ on CIFAR-10 distillation performance.**

compared to training-free sampling methods (e.g., DDIM and DPM-Solver). Additionally, RDD significantly outperforms our training scheme baseline method, RCFD, validating the effectiveness of our approach.

**Results on ImageNet 64×64.** In Table 2, we compare our proposed RDD on ImageNet 64×64 conditional generation with other methods mentioned above. We observe that compared to RCFD, RDD yields FID reduces of 1.47, 0.64, and 0.16 at 1, 2, and 4 steps sampling, while even improving the Inception Score (IS), which also signifies a notable advancement over the PD method. Moreover, RDD demonstrates superior performance with a reduced number of sampling steps, the same as on the CIFAR-10 dataset. Notably, our RDD method at 4 sampling steps even greatly surpasses DDIM at 1024 steps with a remarkable 256× increase in sampling speed. Furthermore, RDD achieves comparable generation quality to PD's 8-step sampling with only 2-step sampling and greatly improves the IS score, effectively reducing the number of sampling steps by 4×. These experimental results underscore the superior or equivalent performance achieved by RDD in fewer sampling steps compared to training-free sampling methods DDIM. Additionally, RDD significantly outperforms our training scheme baseline method, RCFD, validating the effectiveness of our approach on large dataset.

**Visual results comparison.** In Fig. 5 we visualize some of the generated results on ImageNet 64×64. Among them, our method RDD is superior to PD and RCFD in generating details and color. This shows that our method can further improve the effect of distillation and improve the quality of image generation.

## 5.3 Ablation Study and Parameter Analysis

We conduct thorough ablation experiments of our proposed RDD on CIFAR-10 unconditional generation task. For the ablation study of different loss terms, we report the final performance of the 1-step student model for better comparison. For all other experiments, we select the 8-step basic model distilled by PD as the teacher model and report the performance of the 4-step student model.

**Impact of loss terms.** As shown in Table 3, we evaluate the contribution of each distillation loss component. The baseline loss $\mathcal{L}_{CFD}$ greatly enhances the PD framework, proving the effectiveness of feature alignment. Subsequently, incorporating the intra-image relational loss $\mathcal{L}_{II\_P2P}$, the intra-sample relational loss $\mathcal{L}_{IS\_P2P}$ and the memory-based relational loss $\mathcal{L}_{M\_P2P}$ results in additional gains of 0.47, 0.57 and 0.45, respectively, over $\mathcal{L}_{CFD}$. These results

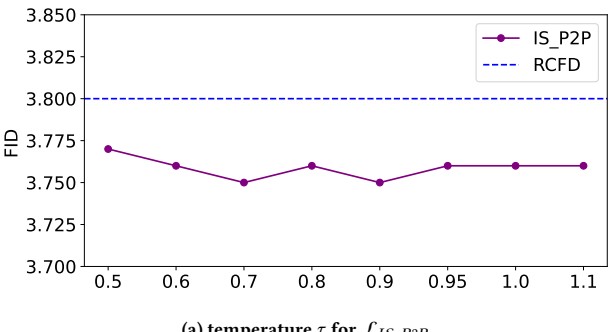

(a) temperature $\tau$ for $\mathcal{L}_{IS\_P2P}$

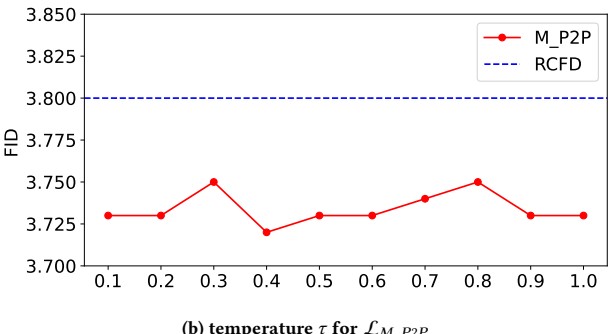

(b) temperature $\tau$ for $\mathcal{L}_{M\_P2P}$

**Figure 7: Impact of (a) temperature $\tau$ for $\mathcal{L}_{IS\_P2P}$ and (b) temperature $\tau$ for $\mathcal{L}_{M\_P2P}$ on CIFAR-10 distillation performance.**

highlight that while $\mathcal{L}_{CFD}$ substantially improves the effectiveness of the PD framework and achieves notable results, our proposed methods can further enhance the generation quality of the diffusion model individually. The significant improvements achieved by each method underscore their effectiveness. It also proves that our proposed intra-sample distillation surpasses intra-image distillation due to its broader sample interaction. Finally, by combining $\mathcal{L}_{IS\_P2P}$ and $\mathcal{L}_{M\_P2P}$, we attain a further improvement of 0.76 over the baseline $\mathcal{L}_{CFD}$, surpassing the results obtained in RCFD. This demonstrates the effectiveness of our proposed methods and their capability to significantly enhance the performance of the diffusion model.

**Impact of batch size for our proposed IS_P2P.** As shown in Fig. 6, we observed that as the batch size increases, IS_P2P yields more pronounced performance improvements albeit at the cost of significantly heightened memory consumption. This underscores the notion that a larger batch size facilitates the student model in learning relationship features from a broader range of samples during the distillation process, thereby enhancing model performance. However, the exponential growth in memory usage poses a considerable challenge in setting an optimal batch size.

**Impact of temperature $\tau$ in loss.** Temperature $\tau$ is utilized in Eq. 6 and Eq. 9 to adjust the distribution for relational knowledge distillation (KD), thereby enhancing performance. A higher temperature $\tau$ results in a smoother distribution. In Fig. 7a and Fig. 7b, we explore the impact of $\tau$ on $\mathcal{L}_{IS\_P2P}$ and $\mathcal{L}_{M\_P2P}$ and compare it with RCFD. Remarkably, both $\mathcal{L}_{IS\_P2P}$ and $\mathcal{L}_{M\_P2P}$ exhibit robustness across different $\tau$ values, with all results surpassing

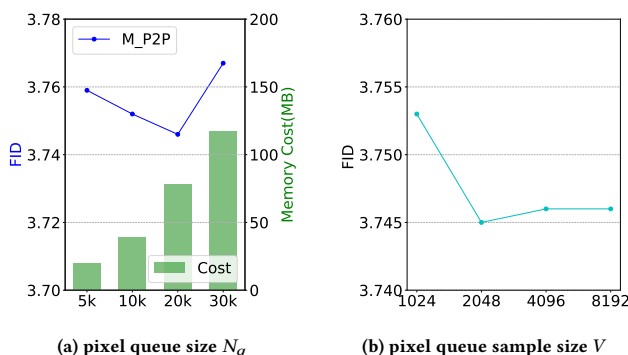

(a) pixel queue size $N_q$          (b) pixel queue sample size $V$

**Figure 8: Impact of (a) pixel queue size $N_q$ and (b) pixel queue sample size $V$ on CIFAR-10 distillation performance.**

those of RCFD. This indicates that our method does not heavily rely on meticulously chosen $\tau$ values to achieve superior distillation outcomes. Specifically, the optimal $\tau$ for $\mathcal{L}_{IS\_P2P}$ is found to be 0.7 and 0.9, while for $\mathcal{L}_{M\_P2P}$, the optimal $\tau$ is 0.4. However, even without fine-tuning $\tau$ in our experiment settings, our method still outperforms RCFD, underscoring its robustness and effectiveness.

**Impact of pixel queue size $N_q$.** We investigate the impact of memory sizes $N_q$ of the pixel queue. As depicted in Fig. 8a, the distillation performance improves as the pixel queue size increases within a certain range, reaching optimal performance before declining beyond a threshold. This phenomenon can be attributed to the fact that within a certain range, a larger pixel queue stores a richer variety of contrastive embeddings, enabling the capture of more relationships between features during distillation. However, when the pixel queue becomes excessively large, it stores an abundance of redundant or irrelevant feature embeddings, leading to learning difficulties and suboptimal performance. It is worth mentioning that the memory cost of our online queue is very small compared to directly increasing the training batch size. Additionally, the results suggest that the distillation performance may saturate at a certain memory capacity, indicating that there is an optimal balance to be struck between memory size and distillation effectiveness. And the optimal $N_q$ for performance is 20k.

**Impact of sampling size $V$ in pixel queue.** As shown in Fig. 8b, we examined the impact of the number of contrastive embeddings $V$ used to calculate the pixel similarity matrix. Our findings indicate that as $V$ increases, the performance of distillation gradually improves until reaching saturation. This observation suggests that a larger number of samples can introduce more feature relationships into the distillation process, facilitating the creation of a more complex pixel similarity matrix. This, in turn, enhances the learning process of the student models. Notably, the hyperparameter $V$ exhibits low sensitivity, as long as the number of samples is not excessively small, enabling the attainment of significantly improved distillation effects. And the optimal $V$ is 2048.

**Impact of loss weights coefficients $\alpha$ and $\beta$.** We investigated the impact of $\alpha$ and $\beta$ in Eq. 10 for $\mathcal{L}_{IS\_P2P}$ and $\mathcal{L}_{M\_P2P}$, respectively. As illustrated in Fig. 9a and Fig. 9b, we observed that both $\mathcal{L}_{IS\_P2P}$ and $\mathcal{L}_{M\_P2P}$ exhibit robustness across different values of $\alpha$ and $\beta$, with all results outperforming RCFD. These findings

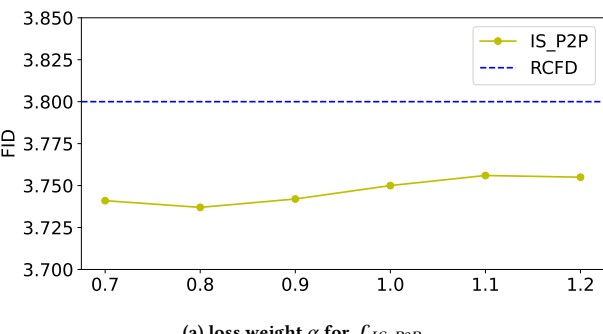

(a) loss weight $\alpha$ for $\mathcal{L}_{IS\_P2P}$

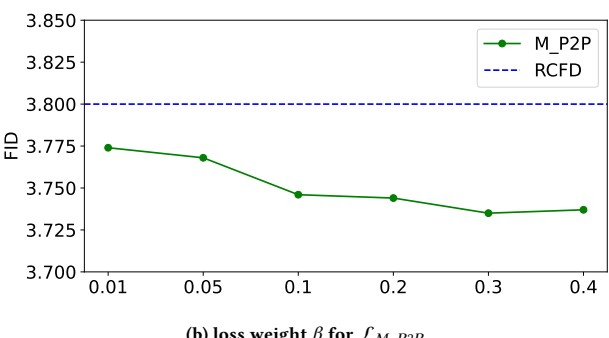

(b) loss weight $\beta$ for $\mathcal{L}_{M\_P2P}$

**Figure 9: Impact of (a) $\alpha$ for $\mathcal{L}_{IS\_P2P}$ and (b) $\beta$ for $\mathcal{L}_{M\_P2P}$ on CIFAR-10 distillation performance.**

indicate that our method does not heavily rely on carefully selected weight coefficients in the total loss $\mathcal{L}_{RDD}$. Notably, the optimal $\alpha$ for $\mathcal{L}_{IS\_P2P}$ is found to be 0.8, while the optimal $\beta$ for $\mathcal{L}_{M\_P2P}$ is 0.3. However, even without fine-tuning $\alpha$ and $\beta$ in our experimental settings, our method still surpasses RCFD, underscoring its robustness and effectiveness.

## 6 CONCLUSION

This paper presents a novel diffusion-specialized distillation method called Relational Diffusion Distillation which introduces intra-sample relation and an online queue to capture broader pixel correlations, greatly enhancing the performance of the progressive distillation framework. Compared to previous methods PD and RCFD, our method helps students learn spatial information in feature maps from the feature extractor and alleviates the deficiency of using only KL divergence. Experiments on CIFAR-10 and ImageNet demonstrate the effectiveness of our Relational Diffusion Distillation. We hope our work can inspire future research to explore better knowledge forms in diffusion model distillation.

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
