# OpenReview forum: "Relational Diffusion Distillation For Efficient Image Generation"
_acmmm.org/ACMMM/2024/Conference — MM2024 Oral_

### Official Review · Reviewer_Lhc1 · 2024-05-24

**Rating:** 4
**Confidence:** 2

**Summary:**

This paper introduces Relational Diffusion Distillation (RDD), a novel method designed to improve the efficiency of diffusion models used in image generation. Diffusion models, while capable of high-quality image generation, suffer from slow inference speeds due to their iterative denoising process. RDD addresses this issue by incorporating cross-sample relationship interaction during the distillation process, which enhances the progressive distillation framework. The method shows significant improvements in both speed and performance, achieving a 1.47 FID decrease and a 256x speed-up compared to existing methods. Extensive experiments on datasets like CIFAR-10 and ImageNet validate the effectiveness of RDD.

**Strengths:**

1. The paper is written in a clear and comprehensible manner, making it accessible to a wide range of readers.
2. The work on accelerating diffusion models is highly relevant and meaningful, particularly for applications on edge devices with limited computational resources.
3.  The proposed method demonstrates significant improvements in both speed and performance, achieving a 256x speed-up and a 1.47 FID decrease compared to state-of-the-art methods.

**Limitations:**

1. Some parts of Figure 4 are not fully rendered, which may hinder the reader's ability to fully understand the visual data presented.
2. The paper primarily focuses on improving existing methods, which may limit its perceived novelty compared to more groundbreaking contributions in the field.
3. Validation has only been conducted on two standard datasets. Is it possible to perform validation in more complex data scenarios?
4. In line 500， does the different pixel embedding sampling mechanism affect the results?

**Suitability:**

2

---

### Official Review · Reviewer_SkSm · 2024-05-26

**Rating:** 4
**Confidence:** 1

**Summary:**

The paper proposes a novel distillation method, Relational Diffusion Distillation (RDD), to enhance the efficiency of diffusion models for image generation. Two distillation strategies in RDD are proposed: (1) Intra-Sample Pixel-to-Pixel Relationship Distillation (IS_P2P) to retain spatial information within feature maps and introduces cross-sample relationship interaction to capture dependencies between image features, and (2) Memory-based Pixel-to-Pixel Relationship Distillation (M_P2P) to address memory inefficiency in IS_P2P.

**Strengths:**

The article is well explained, the motivations are clear and the code is provided.

**Limitations:**

The performance comparison does not include a state-of-the-art method DiffKD [1].

>[1] Knowledge diffusion for distillation. Huang T, Zhang Y, Zheng M, et al.

I am not familiar with this field, but if the author can address my concerns, I would prefer to raise my score.

**Suitability:**

2

---

### Official Review · Reviewer_WXTN · 2024-05-29

**Rating:** 5
**Confidence:** 3

**Summary:**

This paper builds on the recent classifier-based feature distillation - RCFD's success by further integrating feature map spatial information and intra-sample relationship information. The proposed Relational Diffusion Distillation (RDD) method is particularly designed for distilling diffusion models that is composed of two main components: (1) Intra-Sample Pixel-to-Pixel Relationship Distillation, and (2) Memory-based Pixel-to-Pixel Relationship Distillation. Ablation studies have demonstrated the effectiveness of both designs, and results show its better performance compared to the baseline methods including RCFD measured by IS and FID on CIFAR-10 and ImageNet datasets.

**Strengths:**

1. The paper is well-written and easy to follow, with plenty of details explained.
2. The task is well-motivated and is significant to the broader field of accelerating sampling for diffusion models.
3. Helping students to additionally learn spatial information in feature maps is a good supplement to the vanilla RCFD method for boosting performance.
4. Comprehensive experiments on both CIFAR-10 and ImageNet have validated the effectiveness of the method over all baselines, and the ablation study is a strong plus for showing the effectiveness and necessity of each design.

**Limitations:**

While the performance of RDD surpasses the main baseline method (RCFD), the improvement is relatively modest, particularly in contrast to the bold claims in the abstract, which state that "our proposed RDD leads to a 1.47 FID decrease and a 256x speed-up compared to state-of-the-art diffusion distillation methods." This statement raises high expectations regarding its practical significance.

However, as evidenced in Tables 1 and 2:

1. The 1.47 FID decrease is observed only in the most advantageous scenario when tested on ImageNet with 1 sampling step. When using 4 sampling steps on the same dataset, the FID decrease drops to 0.16. Additionally, the FID improvements are not significant for CIFAR-10, and IS performance occasionally falls below that of the baseline method.
2. The claimed 256x speed-up is in comparison to DDIM, an achievement also realized by the baseline RCFD method. Thus, asserting that RDD achieves a 256x speed-up compared to state-of-the-art diffusion distillation methods can be misleading and an overstatement.

Therefore, it is recommended to revise the claims to be more precise. For example:

1. Our proposed RDD leads to a 1.47 FID decrease when tested on ImageNet using 1 sampling step.
2. Similar to RCFD, our proposed RDD also achieves a 256x speed-up compared to DDIM.

**Suitability:**

3

---

### Meta-Review · Area_Chair_oXaD · 2024-06-29

**Recommendation:** Accept (Oral)
**Confidence:** 5

**Metareview:**

This paper proposes Relational Diffusion Distillation, a novel distillation method for diffusion models that introduces cross-sample relationship interaction, enhancing the distillation framework and achieving superior results in experiments on multiple datasets. Overall, the paper is technically solid and will have high impact on AIGC research. Given the resolution of raised concerns and the unanimous positive reviews, the paper is accepted for publication.